# Model-based inference of synaptic plasticity rules

**Yash Mehta**[1,2]    **Danil Tyulmankov**[3,4]    **Adithya E. Rajagopalan**[1,5]

**Glenn C. Turner**[1]    **James E. Fitzgerald**[1,6*]    **Jan Funke**[1*]

[1]Janelia Research Campus, Howard Hughes Medical Institute
[2]Department of Cognitive Science, Johns Hopkins University
[3]Center for Theoretical Neuroscience, Columbia University
[4]Viterbi School of Engineering, University of Southern California
[5]Center for Neural Science, New York University
[6]Department of Neurobiology, Northwestern University

## Abstract

Inferring the synaptic plasticity rules that govern learning in the brain is a key challenge in neuroscience. We present a novel computational method to infer these rules from experimental data, applicable to both neural and behavioral data. Our approach approximates plasticity rules using a parameterized function, employing either truncated Taylor series for theoretical interpretability or multilayer perceptrons. These plasticity parameters are optimized via gradient descent over entire trajectories to align closely with observed neural activity or behavioral learning dynamics. This method can uncover complex rules that induce long nonlinear time dependencies, particularly involving factors like postsynaptic activity and current synaptic weights. We validate our approach through simulations, successfully recovering established rules such as Oja's, as well as more intricate plasticity rules with reward-modulated terms. We assess the robustness of our technique to noise and apply it to behavioral data from *Drosophila* in a probabilistic reward-learning experiment. Notably, our findings reveal an active forgetting component in reward learning in flies, improving predictive accuracy over previous models. This modeling framework offers a promising new avenue for elucidating the computational principles of synaptic plasticity and learning in the brain.

## 1 Introduction

Synaptic plasticity, the ability of synapses to change their strength, is a key neural mechanism underlying learning and memory in the brain. These synaptic updates are driven by neuronal activity, and they in turn modify the dynamics of neural circuits. Advances in neuroscience have enabled the recording of neuronal activity on an unprecedented scale (Steinmetz et al., 2018; Vanwalleghem et al., 2018; Zhang et al., 2023), and connectome data for various organisms is becoming increasingly available (Winding et al., 2023; Takemura et al., 2023; Hildebrand et al., 2017; Scheffer et al., 2020). However, the inaccessibility of direct large-scale recordings of synaptic dynamics leaves the identification of biological learning rules an open challenge. Existing neuroscience literature (Citri & Malenka, 2008; Morrison et al., 2008) suggests that synaptic changes are functions of local variables such as presynaptic activity, postsynaptic activity, and current synaptic weight, as well as a global reward signal. Uncovering the specific form of this function in different brain circuits promises profound biological insights and holds practical significance for developing more biologically plausible learning algorithms for AI, particularly with neuromorphic implementations (Zenke & Neftci, 2021).

---

*Joint senior authors

38th Conference on Neural Information Processing Systems (NeurIPS 2024).

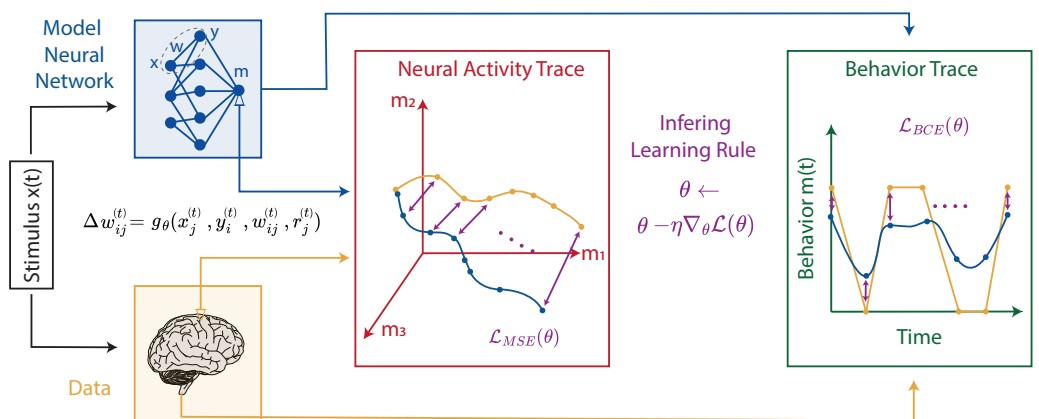

Figure 1: Schematic overview of the proposed method. Animal-derived time-series data (yellow) and a plasticity-regulated *in silico* model (blue) generate trajectories $\boldsymbol{o}^{(t)}$ and $\boldsymbol{m}^{(t)}$. A loss function quantifies trajectory mismatch to produce a gradient, enabling the inference of the synaptic plasticity rule $g_\theta$.

In this paper, we introduce a gradient-based method for inferring synaptic plasticity rules. Our method optimizes parameterized plasticity rules to align with either neural and behavioral data, thereby elucidating the mechanisms governing synaptic changes in biological systems. We utilize interpretable models of plasticity, allowing direct comparisons with existing biological theories and addressing specific questions, such as the role of weight decay in synaptic plasticity or postsynaptic dependence. We validate our approach for recovering plasticity rules using synthetic neural activity or behavior[2]. Finally, applying our model to behavioral data from fruit flies, we uncover an active forgetting mechanism in the neural circuitry underlying decision making. This readily adaptable modeling framework offers new opportunities for exploring the core mechanisms behind learning and memory processes in a variety of experimental paradigms.

## 2 Method overview

Our goal is to infer the synaptic plasticity function by examining neural activity or behavioral trajectories from an animal learning about its environment. Specifically, we aim to find a function that prescribes changes in synaptic weights based on relevant biological variables. For simplicity, we consider a model with plasticity localized to a single layer of a neural network:

$$\boldsymbol{y}^{(t)} = \text{sigmoid}\left(W^{(t)}\boldsymbol{x}^{(t)}\right), \tag{1}$$

where the vector $\boldsymbol{x}^{(t)}$ represents the input to the plastic layer (Figure 1, "stimulus") and $\boldsymbol{y}^{(t)}$ is the resulting postsynaptic neuron activity at time $t$. The synaptic weight matrix $W^{(t)}$ is updated at each time step based on a parameterized biologically plausible plasticity function $g_\theta$. The change in synaptic weight between neurons $i$ and $j$ is given by

$$\Delta w_{ij}^{(t)} = g_\theta\left(x_j^{(t)}, y_i^{(t)}, w_{ij}^{(t)}, r^{(t)}\right), \tag{2}$$

where $\theta$ are the (trainable) parameters of the function, $x_j^{(t)}$ is the presynaptic neural activity, $y_i^{(t)}$ the postsynaptic activity, $w_{ij}^{(t)}$ is the current synaptic weight between neurons $i$ and $j$, and $r^{(t)}$ is a global reward signal that influences all synaptic connections. However, it may be the case that we do not have direct access to the neuronal firing rates $\boldsymbol{y}^{(t)}$. We therefore further define a (fixed) readout function $f$ that determines the observable variables $\boldsymbol{m}^{(t)}$ of the network, given by

$$\boldsymbol{m}^{(t)} = f(\boldsymbol{y}^{(t)}). \tag{3}$$

---

[2]https://github.com/yashsmehta/MetaLearnPlasticity

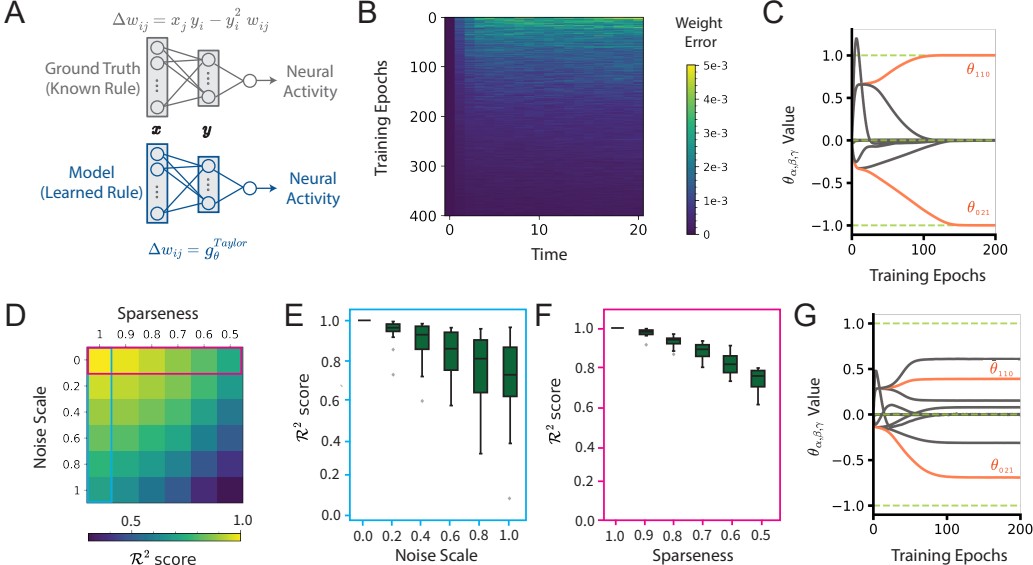

Figure 2: Recovery of Oja's plasticity rule from simulated neural activity. (A) Schematic of the models used to simulate neural activity and infer plasticity. (B) Mean-squared difference between ground-truth and model synaptic weight trajectories over time (horizontal axis) over the course of training epochs (vertical axis). (C) The evolution of $\theta$ during training. Coefficients $\theta_{110}$ and $\theta_{021}$, corresponding to Oja's rule values $(1, -1)$, are highlighted in orange. (D) $R^2$ scores over weights, under varying noise and sparsity conditions in neural data. (E, F) Boxplots of distributions, across 50 seeds, corresponding to the first column (E) and row (F) in (D). (G) The evolution of learning rule coefficients over the course of training showing inaccurate $\theta$ recovery under high noise and sparsity conditions.

In the context of neural activity fitting, the readout is a subset of $\boldsymbol{y}^{(t)}$, whereas for behavioral models the readout aggregates $\boldsymbol{y}^{(t)}$ to yield the probability of a specific action. We introduce our specific choices for readout functions in the following sections.

We use stochastic gradient descent (Kingma & Ba, 2014) to optimize the parameters $\theta$ of the plasticity rule $g_\theta$. At each iteration, we use the model (Equation 1-3) to generate a length-$T$ trajectory $\boldsymbol{m}^{(1)}, \dots, \boldsymbol{m}^{(T)}$ (Figure 1, blue traces), driven by input stimuli $\boldsymbol{x}^{(1)}, \dots, \boldsymbol{x}^{(T)}$ (Figure 1, black box). We then use backpropagation through time to compute the gradient of the loss (Figure 1, purple) between the model trajectory and the corresponding experimental observations $\boldsymbol{o}^{(1)}, \dots, \boldsymbol{o}^{(T)}$ (Figure 1, orange) generated using the same input stimulus:

$$\mathcal{L}(\theta) = \frac{1}{T} \sum_{t=1}^{T} \ell(\boldsymbol{m}^{(t)}, \boldsymbol{o}^{(t)}), \tag{4}$$

where the choice of $\ell$ depends on the particular modeling scenario, specified in the following sections. In practice, Equation 4 may also be summed over multiple trajectories to generate a mini-batch.

## 3 Inferring a plasticity rule from neural activity

To validate our approach on neural activity, we generate synthetic neural trajectories of observed outputs $\boldsymbol{o}^{(t)}$ from a single-layer feedforward network that undergoes synaptic plasticity according to the well-known Oja's rule (Oja, 1982). At each timestep, the weight updates depend on pre- and post-synaptic neuronal activity, as well as the strength of the synapse itself (Figure 2A, top),

$$\Delta w_{ij} = x_j y_i - y_i^2 w_{ij}, \tag{5}$$

where we omit the time index $t$ for brevity (see subsection A.3 for details). To infer the plasticity rule, we use a model network with an architecture identical to the ground-truth network (Figure 2A, bottom). While the exact circuit architecture may not always be known in biological data (particularly when studying mammalian brains), this assumption was made to show that our approach accurately infers plasticity rules in a scenario where generative and predictive circuit architectures were matched. Increasingly available connectomic information in biological systems can be used to design sufficiently accurate model architectures for our approach to work, as we will show later on in this paper (Bentley et al., 2016; Hildebrand et al., 2017; Scheffer et al., 2020). However, it is to be noted that connectomes can miss important information relevant to plasticity and may cause mismatches between model and generative architectures that could lead the model to incorrect outcomes (Liang & Brinkman, 2024). This should be kept in mind when interpreting the plasticity rules estimated by our approach.

Following previous work (Confavreux et al., 2020), we parameterize the model's plasticity function with a truncated Taylor series,

$$g_\theta^{\text{Taylor}} = \sum_{\alpha, \beta, \gamma=0}^{2} \theta_{\alpha\beta\gamma} x_i^\alpha y_j^\beta w_{ij}^\gamma, \tag{6}$$

where the coefficients $\theta_{\alpha\beta\gamma}$ are learned. Note that Oja's rule can be represented within this family of plasticity rules by setting $\theta_{110} = 1$, $\theta_{021} = -1$, and all others to zero. Finally, we compute the loss as the mean squared error (MSE) between the neural trajectories produced by the ground truth network and the model:

$$\ell_{\text{MSE}}(\boldsymbol{m}^{(t)}, \boldsymbol{o}^{(t)}) = ||\boldsymbol{o}^{(t)} - \boldsymbol{m}^{(t)}||^2, \tag{7}$$

where we let $\boldsymbol{m}^{(t)} = \boldsymbol{y}^{(t)}$, assuming all neurons in the circuit are recorded (but see next section for analysis of sparse recordings).

### 3.1 Recovering Oja's rule

Despite the fact that the model is optimized using *neuronal* trajectories, the error of the *synaptic* weight trajectories decreases over the course of training (Figure 2B), indicating that the model successfully learns to approximate the ground-truth plasticity rule. More explicitly examining the coefficients $\theta_{\alpha\beta\gamma}$ over the course of training illustrates the recovery of Oja's rule as $\theta_{110}$ and $\theta_{021}$ approach 1 and $-1$, respectively, and all others go to zero (Figure 2C).

To evaluate the robustness of our method, we assess how both noise and sparsity affect the model's performance (Figure 2D). We first consider the case where all neurons in the circuit are recorded, and we vary the degree of additive Gaussian noise in the recorded neurons. We find that the model's performance ($R^2$ score between the ground-truth and model synaptic weight trajectories calculated on a separate held-out test set) decreases with increasing noise variance (Figure 2E). To simulate varying sparsity levels, we consider the readout $\boldsymbol{m}^{(t)} = f(\boldsymbol{y}^{(t)}) = (y_{k_1}^{(t)}, \ldots, y_{k_n}^{(t)})$ to be the activity of a random subset $n$ of all $N$ postsynaptic neurons $\boldsymbol{y}^{(t)}$, and we use the corresponding subset $\widetilde{\boldsymbol{o}}^{(t)} = (o_{k_1}^{(t)}, \ldots, o_{k_n}^{(t)})$ of recorded neurons from the ground-truth network in Equation 7 to optimize the plasticity rule. Our model maintains a high level of accuracy even when data is available from only 50% of the neurons (Figure 2F). This resilience to sparsity and noise is beneficial given that experimental neural recordings often suffer from these issues. However, we note that the model struggles to learn a sparse set of parameters for the plasticity rule when faced with both high recording sparsity and noise. The evolution of the plasticity parameters during training in this case is illustrated in Figure 2G. Together, these results show that our approach can accurately infer learning rules from neural trajectories in a wide array of recording conditions.

## 4 Inferring plasticity rules from behavior

Our approach can also be applied to behavioral data. This is particularly important because behavioral experiments are more widely available and easier to conduct than those that directly measure neural activity. We first validate the method on simulated behavior, mimicking decision-making experiments in which animals are presented with a series of stimuli that they choose to accept or reject. The animals' choices result in rewards and subsequent synaptic changes at behaviorally relevant synapses.

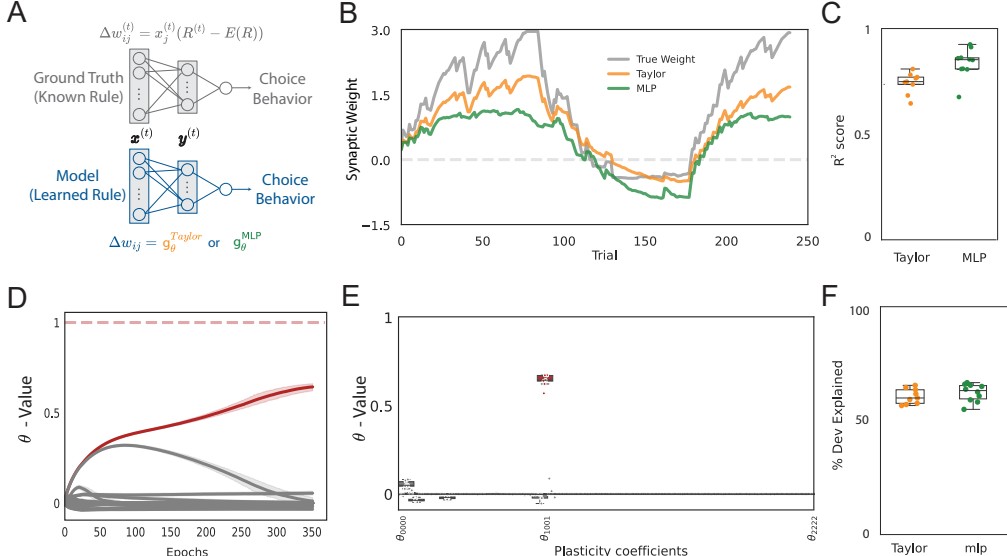

Figure 3: Recovery of a reward-based plasticity rule from simulated behavior. (A) Schematic of the models used to stimulate behavior and infer plasticity rules. (B) The evolution of the weight of a single synapse, trained with $g_\theta^{\text{Taylor}}$ and $g_\theta^{\text{MLP}}$, compared against weight from a known reward-based update rule. (C) $R^2$ distributions on the weights across 10 seeds, corresponding to varied weight initializations and stimulus encodings. (D) The evolution of $\theta$ during training, with $\theta_{110}$, corresponding to ground truth rule (value = 1), highlighted in red. (E) Distribution of final inferred $\theta$ values across seeds, showing accurate identification of the relevant term from the ground truth learning rule. (F) The goodness of fit between ground truth behavior and model predictions plotted as the percent deviance explained.

For this proof-of-principle, our ground-truth network architecture (Figure 3A, top) is inspired by recent studies that have successfully mapped observed behaviors to plasticity rules in the mushroom body (MB), the learning and memory center of the fruit fly *Drosophila melanogaster* (Li et al., 2020; Modi et al., 2020; Davis, 2023). Our neural network's three layer structure mimics the MB's neural architecture (see subsection A.4 for details). The readout $\boldsymbol{m}^{(t)}$ is a series of binary decisions to either "accept" or "reject" the presented stimulus based on the average activity of the output layer. A probabilistic binary reward $\mathcal{R} \in \{0, 1\}$ is provided based on the choice. The reward signal is common to all synapses, which could be interpreted as a global neuromodulatory signal like dopamine. This reward leads to changes in the plastic weights of the network, determined by the underlying synaptic plasticity rule.

Plasticity occurs exclusively between the input and output layers. We simulate a covariance-based learning rule (Loewenstein & Seung, 2006) known from previous experiments (Rajagopalan et al., 2023). The change in synaptic weight $\Delta w_{ij}$ is determined by the presynaptic input $x_j$, and a global reward signal $r$. This reward signal is the deviation of the actual reward $\mathcal{R}$ from its expected value $\mathbb{E}[\mathcal{R}]$, which we calculate as a moving average over the last 10 trials. We neglect hypothetical dependencies on $y_i$ because they are known to not impact reward learning in the fly mushroom body (although see also Table 1 and Appendix Table 3 for experiments with alternative plasticity rules):

$$\Delta w_{ij} = x_j r = x_j(\mathcal{R} - \mathbb{E}[\mathcal{R}]). \tag{8}$$

We model a plastic layer of neural connections that gives rise to learned behavior. The synaptic weights of the model are initialized randomly, reflecting the fact that the initial synaptic configurations are usually unknown *a priori* in real-world biological systems. We consider a plasticity function parameterized through either a truncated Taylor series or a multilayer perceptron (MLP):

$$g_\theta^{\text{Taylor}} = \sum_{\alpha,\beta,\gamma,\delta=0}^{2} \theta_{\alpha\beta\gamma\delta} x_j^\alpha y_i^\beta w_{ij}^\gamma r^\delta \quad \text{or} \quad g_\theta^{\text{MLP}} = \text{MLP}_\theta(x_j, y_i, w_{ij}, r), \tag{9}$$

Table 1: Assessment of various reward-based plasticity rules: $R^2$ scores for weight and individual neural activity trajectories, and the percentage of deviance explained for behavior. Refer to Appendix Table 3 for a comprehensive list of simulated plasticity rules.

| Plasticity Rule $\Delta w_{ij}$ | MLP | | | Taylor | | |
|---|---|---|---|---|---|---|
| | $R^2$ Weights | $R^2$ Activity | % Deviance | $R^2$ Weights | $R^2$ Activity | % Deviance |
| $x_j r$ | 0.85 | 0.96 | 64.76 | 0.78 | 0.94 | 61.91 |
| $x_j r^2 - 0.05 y_i$ | 0.97 | 0.97 | 34.55 | 0.96 | 0.97 | 34.36 |
| $x_j r - 0.05 w_{ij}$ | 0.87 | 0.91 | 57.01 | 0.70 | 0.86 | 51.80 |
| $x_j r^2 - 0.05 x_j w_{ij} r$ | 0.85 | 0.96 | 53.04 | 0.78 | 0.92 | 51.30 |
| $x_j y_i w_{ij} r - 0.05 r$ | 0.27 | 0.34 | 79.92 | 0.41 | 0.51 | 84.22 |

where the ground truth reward $r$ is used as the reward value in the weight update function. We use Binary Cross-Entropy (BCE) as the loss function, which is proportional to the model's negative log-likelihood function, to quantify the difference between the observed decisions and the model's predicted probabilities of accepting a stimulus.

$$\ell_{\text{BCE}}(\boldsymbol{m}^{(t)}, \boldsymbol{o}^{(t)}) = -\boldsymbol{o}^{(t)} \log(\boldsymbol{m}^{(t)}) - (1 - \boldsymbol{o}^{(t)}) \log(1 - \boldsymbol{m}^{(t)}). \tag{10}$$

Crucially, the training data only consists of these binary decisions (accept or reject), without direct access to the underlying synaptic weights or neural activity.

## 4.1 Recovering reward-based plasticity from behavior

Figure 3B presents the weight dynamics of three networks: the ground-truth synaptic update mechanism, as well as those fitted with an MLP or a Taylor series. Both the ground-truth network and our model use an architecture with 2 input neurons, 10 neurons in the hidden layer, and 1 output neuron with a trajectory length of 240 time steps (see Appendix subsection A.4). Our evaluation metrics – high $R^2$ values for both synaptic weights and neural activity – affirm the robustness of our models in capturing the observed data (Figure 3C). The method accurately discerns the plasticity coefficient of the ground truth rule (Figure 3D,E), albeit with a reduced magnitude. The model also does a good job at explaining the observed behavior (Figure 3F), where we use the percent deviance explained (see Appendix subsection A.4) as the performance metric.

We also consider alternative plasticity rules in the ground-truth network. Table 1 summarizes the recoverability of various reward-based plasticity rules for both MLP and Taylor series frameworks, with results averaged over 3 random seeds. Note that solely reward-based rules (without $\mathbb{E}[\mathcal{R}]$ or $w$) are strictly potentiating, as they lack the capacity for bidirectional plasticity. This unidirectional potentiation ultimately results in the saturation of the sigmoidal non-linearity. Therefore, it is possible to simultaneously observe high $R^2$ values for neural activities with low $R^2$ values for weight trajectories.

We further investigate the scalability of our method by varying the length of the observed trajectory and the number of neurons in the hidden layer (see Table 2). The model's goodness-of-fit generally improved with longer simulations, likely due to more data points for inferring the plasticity rule. However, $R^2$ values for activity and weights peaked before declining, suggesting potential overfitting on very long trajectories. Model performance remained consistent when scaling to larger hidden layers, assuming the same plasticity rule is shared by all synapses.

## 5 Application: inferring plasticity in the fruit fly

In extending our results to biological data, we explore its applicability to the decision-making behavior in *Drosophila melanogaster* that inspired our simulated behavior results. Recent research (Rajagopalan et al., 2023) employed logistic regression to infer learning rules governing synaptic plasticity in the mushroom body, identifying a rule that incorporates the difference between received and expected reward information when modulating synaptic plasticity. However, logistic

Table 2: Scalability analysis with respect to trajectory length (with a hidden layer size of 10) and hidden layer size (with a trajectory length of 240), assuming the ground-truth learning rule of $\Delta w_{ij} = x_j r = x_j \left( \mathcal{R} - \mathbb{E}[\mathcal{R}] \right)$ and using the Taylor series parameterization. Results are averaged over three runs with different random seeds.

| | Trajectory Length | | | | | | | Hidden Layer Size | | | | |
|---|---|---|---|---|---|---|---|---|---|---|---|---|
| | 30 | 60 | 120 | 240 | 480 | 960 | 1920 | 10 | 50 | 100 | 500 | 1000 |
| $R^2$ Weights | 0.79 | 0.74 | 0.70 | 0.78 | 0.72 | 0.53 | 0.64 | 0.78 | 0.75 | 0.79 | 0.79 | 0.79 |
| $R^2$ Activity | 0.92 | 0.91 | 0.91 | 0.94 | 0.95 | 0.87 | 0.92 | 0.94 | 0.94 | 0.95 | 0.95 | 0.95 |
| % Deviance | 39.52 | 40.14 | 48.06 | 61.91 | 76.90 | 73.78 | 79.66 | 61.91 | 62.29 | 62.25 | 62.27 | 62.26 |

regression cannot be used to infer plasticity rules that incorporate recurrent temporal dependencies, such as those that depend on current synaptic weights. Our method offers a more general approach. Specifically, we apply our model to behavioral data obtained from flies engaged in a two-alternative choice task, as outlined in Figure 4A. This allows us to investigate two key questions concerning the influence of synaptic weight on the plasticity rules governing the mushroom body.

## 5.1 Experimental setup and details

In the experimental setup, individual flies are placed in a symmetrical Y-arena where they are presented with a choice between two odor cues. Each trial starts with the fly in an arm filled with clean air (Fig. 4A, left). The remaining two arms are randomly filled with two different odors, and the fly was free to navigate between the three arms. When the fly enters the 'reward zone' at the end of an odorized arm, a choice was considered to have been made (Fig. 4A, right). Rewards are then dispensed probabilistically, based on the odor chosen. For model fitting, we use data from 18 flies, each subjected to a protocol that mirrors the trial and block structures in the simulated experiments presented previously. Over time, flies consistently showed a preference for the odor associated with a higher probability of reward, and this preference adapted to changes in the relative value of the options (Fig. 4B; example fly (Rajagopalan et al., 2023)).

## 5.2 Plasticity in the fruit fly includes a synaptic weight decay

Existing behavioral studies in fruit flies have shown that these insects can forget learned associations between stimuli and rewards over time (Shuai et al., 2015; Aso & Rubin, 2016; Berry et al., 2018; Gkanias et al., 2022). One prevailing hypothesis attributes this forgetting to homeostatic adjustments in synaptic strength within the mushroom body (Davis & Zhong, 2017; Davis, 2023). However, earlier statistical approaches aimed at estimating the underlying synaptic plasticity rule present in the mushroom body were unable to account for recurrent dependencies such as synapse strength (Rajagopalan et al., 2023). Here we explore two types of plasticity rules: one based solely on reward and presynaptic activity, and another that incorporates a term dependent on current synaptic weight - $w_{ij}$ ($\theta_{001}$). Both rule types allocate significant positive weights to a term representing the product of presynaptic activity and reward (Fig. 4C, gray). Our results indicate that the model with a weight-dependent term offers a better fit to observed fly behavior (Wilcoxon signed-rank test: $p = 5 \times 10^{-5}$; Fig. 4D), whereas the model without it matched the performance reported in Rajagopalan et al. (2023). Intriguingly, our analysis additionally reveals that the inferred learning rule assigns a negative value to the weight-dependent term (Fig. 4C). This finding also held in a validation experiment that assessed how well the inferred plasticity rule generalizes to unseen biological data (Appendix subsection A.6). This negative sign aligns with the hypothesis that a weight-dependent decay mechanism operates at these synapses. The relative-magnitude of this decay term compared to the positive learning-related terms suggests that forgetting happens over a slightly longer-time scale than learning, in agreement with observed time-scales of forgetting in behavioral experiments Shuai et al. (2015); Davis & Zhong (2017) .

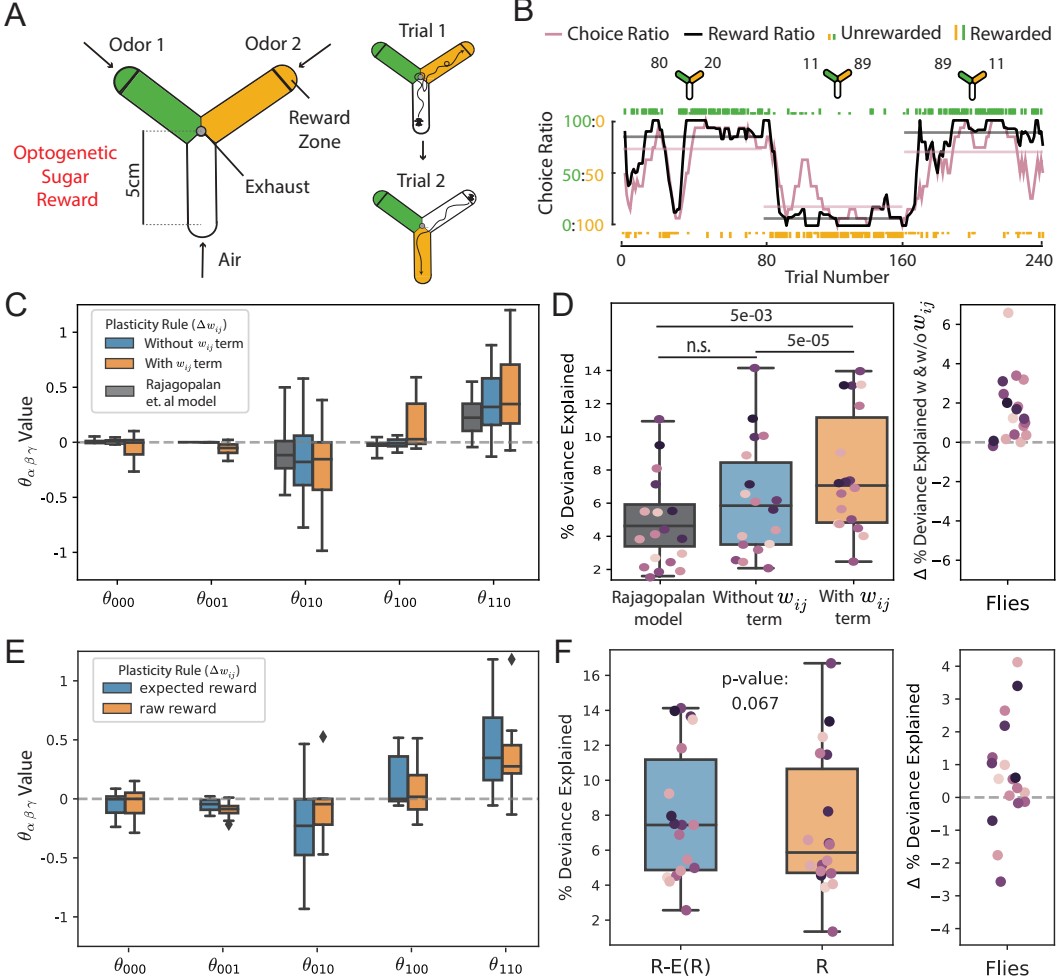

Figure 4: Inferring principles of plasticity in the fruit fly. (A) Schematic of the experimental setup used to study two-alternative choice behavior in flies. *Left* Design of arena showing odor entry ports and location of the reward zones. *Right* Description of the trial structure, showing two example trials. (B) The behavior of an example fly in the task. *Top* Schematics indicate the reward baiting probabilities for each odor in the three blocks. *Bottom* Individual odor choices are denoted by rasters, tall rasters - rewarded choices, short rasters - unrewarded choices. Curves show 10-trial averaged choice (red) and reward (black) ratios, and horizontal lines the corresponding averages over the 80-trial blocks. (C) Final inferred $\theta$ value distribution across 18 flies, comparing models with and without a $w_{ij}$ term and the method from Rajagopalan et al. (2023). Plasticity rule terms are as follows: bias - ($\theta_{000}$), $w_{ij}$ - ($\theta_{001}$), $x_j$ - ($\theta_{100}$), $r$ - ($\theta_{010}$), $x_j r$ - ($\theta_{110}$) (D) *Left* Goodness of fit between fly behavior and model predictions plotted as the percent deviance explained (n = 18 flies). *Right* Change in the percent deviance explained calculated by subtracting percent deviance explained of model without a $w_{ij}$ ($\theta_{001}$) term from that of a model with a $w_{ij}$ ($\theta_{001}$) term. (E,F) Same as (C,D), except comparing models that do or don't incorporated reward expectation. Since these models include weight dependence, they cannot be fit using Rajagopalan et al. (2023)'s method.

## 5.3 Incorporating reward expectation provides better fit than reward alone

Rajagopalan and colleagues used reward expectations (defined as the average reward received over the last approximately 'n' trials - see Appendix A.4.2) to generate bidirectional synaptic plasticity. Our discovery of a negative weight-dependent component in the plasticity rule provides an alternate mechanism for bidirectional plasticity, raising the question of whether the neural circuit really needs to calculate reward expectation. Could a plasticity rule incorporating the product of presynaptic activity and absolute reward combine with a weight-dependent homeostatic term to approximate a

plasticity rule that involves reward expectation? To answer this, we contrast two models: one using only the absolute reward and another using reward adjusted by its expectation, both complemented by weight-dependent terms. Our analyses show that adding a weight-dependent term enhances the predictive power of both models (Fig 4E,F). However, the model that also factors in reward expectations provides a superior fit for the majority of flies in the data set (Wilcoxon signed-rank test: $p = 0.067$; Fig 4F). These compelling preliminary findings reaffirm the utility of reward expectations for fly learning, and larger behavioral datasets could increase the statistical significance of the trend. Overall, our model-based inference approach, when applied to fly choice behavior, suggests that synaptic plasticity rules in the mushroom body of fruit flies are more intricate than previously understood. These insights could potentially inspire further experimental work to confirm the roles of weight-dependent homeostatic plasticity and reward expectation in shaping learning rules.

# 6    Related work

Recent work has begun to address the question of understanding computational principles governing synaptic plasticity by developing data-driven frameworks to infer underlying plasticity rules from neural recordings. Lim et al. (2015) infer plasticity rules, as well as the neuronal transfer function, from firing rate distributions before and after repeated viewing of stimuli in a familiarity task. The authors make assumptions about the distribution of firing rates, as well as a first-order functional form of the learning rule. Chen et al. (2023) elaborate on this approach, fitting a plasticity rule by either a Gaussian process or Taylor expansion, either directly to the synaptic weight updates or indirectly through neural activity over the course of learning. Both approaches consider only the difference in synaptic weights before and after learning. In contrast, our approach fits neural firing rate *trajectories* over the course of learning and can be adapted to fit any parameterized plasticity rule.

Other work infers learning rules based on behavior instead. Ashwood et al. (2020) uses a Bayesian framework to fit parameters of learning rules in a rodent decision-making task. The authors explicitly optimize the weight trajectory in addition to parameters of the learning rules, requiring an approximation to the posterior of the weights. Our approach directly optimizes the match between the model and either neural or behavioral data, as defined by a pre-determined loss function. Interestingly, despite this indirect optimization, we see matching in the weight trajectories as well. Rajagopalan et al. (2023) fit plasticity rules in the same fly decision-making task we consider here. They assumed that the learning rule depended only on presynaptic activity and reward, which recasts the problem as logistic regression and permits easy optimization. Our approach allows us to account for arbitrary dependencies, such as on postsynaptic activities and synaptic weight values, and we thereby identify a weight decay term that leads to active forgetting.

Ramesh et al. (2023) also consider optimization of plasticity rules based on neural trajectories. Unlike our approach which uses an explicit loss function, the authors use a generative adversarial (GAN) approach to construct a generator network, endowed with a plasticity rule, to produce neural trajectories that are indistinguishable by a discriminator network from ground-truth trajectories. Although, in principle, this approach can account for arbitrary and unknown noise distributions, it comes at a cost of high compute resources, a need for large amounts of data, and potential training instability – all well-known limitations of GANs. In practice, it is common to make an assumption about the noise model through an appropriately defined loss function (e.g. Gaussian noise for the MSE loss we use here). Importantly, the authors note a degeneracy of plasticity rules – different rules leading to similar neural dynamics. We see similar results, although we interpret this as "sloppiness" (Gutenkunst et al., 2007) – overparameterized models being underconstrained by the data (e.g. functions $x$ and $x^2$ are indistinguishable if the only values of $x$ which are sampled are 0 and 1). We hypothesize that in the infinite data limit the fitted plasticity rules would, in fact, be unique.

Previous work has also considered inferring plasticity rules directly from spiking data (Stevenson & Koerding, 2011; Robinson et al., 2016; Linderman et al., 2014; Wei & Stevenson, 2021) or selecting families of plausible rules in spiking neural network models (Confavreux et al., 2024). Due to the gradient-based nature of our optimization technique, our proposed approach can account for such data by converting spike trains to a rate-based representation by smoothing. Alternatively, black-box optimization techniques such as evolutionary algorithms can be used to circumvent the need

for computing gradients, allowing non-differentiable plasticity rules like spike-timing dependent plasticity to be used as model candidates.

Alternatively, meta-learning techniques (Thrun & Pratt, 2012) can be used to discover synaptic plasticity rules optimized for specific computational tasks (Tyulmankov et al., 2022; Najarro & Risi, 2020; Confavreux et al., 2020; Bengio et al., 1990). The plasticity rules are represented as parameterized functions of pre- and post-synaptic activity and optimized through gradient descent or evolutionary algorithms to produce a desired network output. However, the task may not be well-defined in biological scenarios, and the network's computation may not be known *a priori*. Our method obviates the need for specifying the task, directly inferring plasticity rules from recorded neural activity or behavioral trajectories.

Finally, Nayebi et al. (2020) do not fit parameters of a learning rule at all, but use a classifier to distinguish among four classes of learning rules based on various statistics (e.g. mean, variance) of network observables (e.g. activities, weights). Similarly, Portes et al. (2022) propose a metric for distinguishing between supervised and reinforcement learning algorithms based on changes in neural activity flow fields in a recurrent neural network.

## 7 Limitations and future work

Despite its strengths, our model has several limitations that offer avenues for future research. One such limitation is the lack of complex temporal dependencies in synaptic plasticity, neglecting biological phenomena like metaplasticity (Abraham, 2008). Extending our model to account for such temporal dynamics would increase its biological fidelity. Another issue is the model's "sloppiness" in the solution space; it can fail to identify a unique, sparse solution even with extensive data. As neural recording technologies like Neuropixels (Steinmetz et al., 2021, 2018) and whole-brain imaging (Vanwalleghem et al., 2018) become more advanced, and connectome data for various organisms become increasingly available (Bentley et al., 2016; Hildebrand et al., 2017; Scheffer et al., 2020; Winding et al., 2023), there are exciting opportunities for validating and refining our approach. Incorporating these high-resolution, large-scale datasets into our model is a crucial next step. In particular, future work could focus on scaling our approach to work with large-scale neural recordings and connectomics, offering insights into the spatial organization of plasticity mechanisms. Such refinements will be important when applying our approach to larger and more densely connected brains, such as mammalian ones. Additional considerations for future research include the challenges posed by unknown initial synaptic weights, the potential necessity for exact connectome information, and the adequacy of available behavioral data for model fitting.

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

# A Appendix

## A.1 Glossary

**Backpropagation through time**: An algorithm for training recurrent neural networks by unrolling the network over time steps and applying the standard backpropagation algorithm, allowing the network to learn from sequences of inputs and capture temporal dependencies.

**Binary Cross-Entropy (BCE)**: A loss function commonly used in binary classification problems that measures the difference between predicted probabilities and actual binary labels, encouraging the model to output confident and accurate predictions.

**Connectome**: A comprehensive map of neural connections in the brain, detailing the structural connectivity between neurons or brain regions, providing insights into brain organization and function.

**Covariance-based learning rule**: A learning rule that updates weights based on the covariance between neural activity and reward, adjusting connection strengths to capture statistical relationships in the data and potentially improving the network's performance.

**Deviance explained**: A statistical measure of goodness of fit for generalized linear models, quantifying the proportion of variability in the data that is accounted for by the model compared to a null model.

**Gaussian process**: A statistical model used for regression and probabilistic classification, defining a distribution over functions and allowing for flexible, non-parametric modeling with uncertainty quantification.

**Generative adversarial network (GAN) approach**: A machine learning framework where two neural networks compete to generate realistic data, with one network (the generator) creating synthetic samples and another (the discriminator) trying to distinguish between real and fake data.

**Homeostatic adjustments**: Changes in neural systems that help maintain stability, including mechanisms like synaptic scaling and intrinsic plasticity that regulate neuronal activity and prevent excessive excitation or inhibition.

**Logistic regression**: A statistical method for predicting a binary outcome by modeling the probability of an event occurring as a function of input variables, using the logistic function to transform linear combinations of features into probabilities.

**Synaptic plasticity**: The ability of synapses (connections between neurons) to change their strength over time, forming the basis for learning and memory in the brain.

**Truncated Taylor series**: A mathematical method for approximating functions using polynomial terms, where the series is cut off after a finite number of terms to balance accuracy and computational tractability.

**Meta-learning**: A subfield of machine learning focused on improving the learning process itself, developing algorithms that can learn how to learn and adapt quickly to new tasks or environments.

**Metaplasticity**: Higher-order plasticity where prior synaptic activity influences subsequent plasticity, regulating the threshold and magnitude of future synaptic changes to maintain network stability and optimize learning.

**Multilayer perceptrons (MLPs)**: A type of artificial neural network with multiple layers of nodes, including input, hidden, and output layers, capable of learning complex non-linear relationships in data through backpropagation.

**Mushroom body**: A region in the insect brain involved in learning and memory, particularly important for olfactory learning, sensory integration, and decision-making processes.

**Neuromorphic**: Referring to artificial systems that mimic biological neural systems, often implemented in hardware to achieve brain-like computation with high efficiency and parallelism.

**Neuropixels**: A type of high-density neural recording probe that allows simultaneous recording from hundreds of neurons across multiple brain regions with high spatial and temporal resolution.

**Oja's rule**: A famous local learning rule for neural networks that performs principal component analysis, extracting the dominant features from input data by updating synaptic weights based on the correlation between pre- and post-synaptic activity.

## A.2 Implementation

In experimental contexts, the observed behavioral or neural trajectories $o^{(t)}$ can span extensive time scales, often consisting of thousands of data points. Computing gradients over such long trajectories is computationally demanding. Our framework is implemented in JAX (Bradbury et al., 2018) and is designed to accommodate plasticity modeling across millions of synapses, anticipating future integration with large-scale connectomics datasets. In terms of time, fitting Taylor coefficients on a network with approximately $10^6$ synapses and 1000-time step trajectories takes 2 hours on an NVIDIA H100 GPU.

## A.3 Additional detail: Inferring a plasticity rule from neural activity

To create a ground-truth system with a known plasticity rule, we use a single-layer neural network with 100 input neurons and $N = 1000$ output neurons to generate synthetic data. The output layer employs a sigmoid activation function, chosen for its differentiability and biologically plausible output range of [0, 1]. We generate 50 training trajectories for the network. The input data is sampled from a Gaussian distribution with zero mean and a variance of 0.1, independent of time. A subset of neurons, determined by the sparsity factor, is selected for the readout. The simulation runs over 50 time steps, calculating the neural activity of all output neurons at each step. To ensure numerical stability and prevent exploding gradients, gradient clipping is applied with a threshold of 0.2.

The coefficients $\theta_{\alpha\beta\gamma}$ of the Taylor series expansion representing the plasticity rule are learned, initialized independently and identically distributed (i.i.d.) from a normal distribution with a mean of 0 and a variance of $10^{-4}$. Both the ground-truth and model network weights are initialized using Kaiming initialization from a zero-mean Gaussian distribution. Although these weights are drawn from the same distribution, they are resampled, resulting in different initial values. No regularization is applied during training. The Adam optimizer is used to train the weights of the plasticity model, with default parameters.

## A.4 Additional detail: Inferring plasticity rules from behavior

In the behavioral simulation experiments, the ground truth plasticity rule is denoted as $x_j r$. We use a network with a 2-10-1 architecture and a sigmoid non-linearity. The plasticity MLP has a size of 4-10-1. The default L1 regularization is set to 1e-2, the moving average window is 10, and the input firing mean is 0.75.

Given the discrete nature of the observed behavior and the continuous output of the model, we employ the percent deviance explained as a performance metric. The percent deviance explained measures the model's ability to account for the variability in observed binary choices compared to a null model that assumes no plasticity (i.e., the weights remain at their initial random initialization). It represents the reduction in deviance relative to this null model, expressed as a percentage. Higher values indicate greater log-likelihoods, signifying a superior fit to the observed data.

$$\text{Percent Deviance Explained} = 100 \times \left(1 - \frac{\text{Deviance}_{\text{model}}}{\text{Deviance}_{\text{null}}}\right) \tag{11}$$

Since there are 2 odors, they are encoded in a stimulus vector of dimension 2. Odor 1 corresponds to the first dimension "firing"; for these experiments, we use a value of 0.75, which we call the input firing mean (not 1 to allow the model to differentiate between $x$ and $x^2$ in the plasticity rule). There is Gaussian noise with zero mean and a variance of 0.05 added to account for biological variability in the signal. In one trajectory, traditionally there are 240 trials, consisting of 3 blocks with different reward contingencies for odors in each block. The reward ratios are:

|         | Odor A | Odor B |
|---------|--------|--------|
| Block 1 | 0.2    | 0.8    |
| Block 2 | 0.9    | 0.1    |
| Block 3 | 0.2    | 0.8    |

Updates are performed on a single trajectory, with no batching. In our simulations, we use 18 trajectories for training (matching the size of our previous experimental data) and 7 for evaluation for each seed. For each seed, results are reported as the median. Unless stated otherwise, all reported results are averaged over 3 seeds.

## A.5 Additional experimental parameters

In the following subsections we explore the effect of three factors (regularization, moving average window, input firing mean) on model performance. Only the parameter being tested is varied while the other parameters are held fixed at the previously defined values.

### A.5.1 L1 regularization

We experiment with various values of the L1 regularization penalty applied to the Taylor coefficients. This encourages sparse plasticity solutions and prevents the coefficients from exploding into NaNs due to the learning of positive values that exponentially increase the synaptic weight as the number of time points grows. We do not apply L1 regularization to the MLP parameters. The results of these experiments supports our original choice of L1 regularization level (Fig. 5).

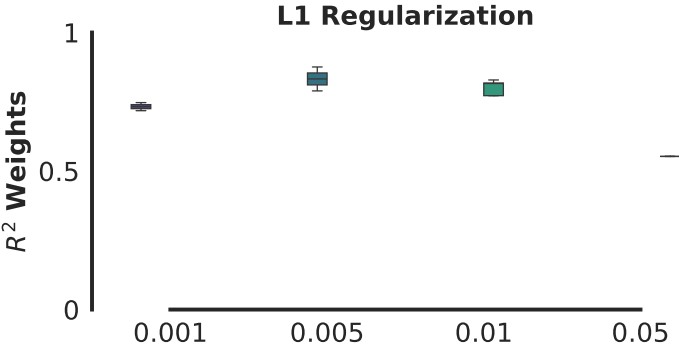

Figure 5: Effect of L1 regularization on $R^2$ of weights for Taylor plasticity rule

### A.5.2 Moving average window

The moving average window refers to the window size used for calculating the expected reward. For example, a moving average window of 10 would take the average of the rewards received over the last 10 trials. Our exploration of window size suggests that smaller windows allow our model to more accurately predict weights in behavioral simulation experiments (Fig. 6). This is to be expected as shorter historical dependencies on past trials reduce the noisiness of the expected reward estimates. Importantly however, our model still robustly identified learning rules when using out default choice of 10 trials which was guided by experimental results from Rajagopalan et al. (2023) that suggested that choice on a given trial was mediated by 10 past trials.

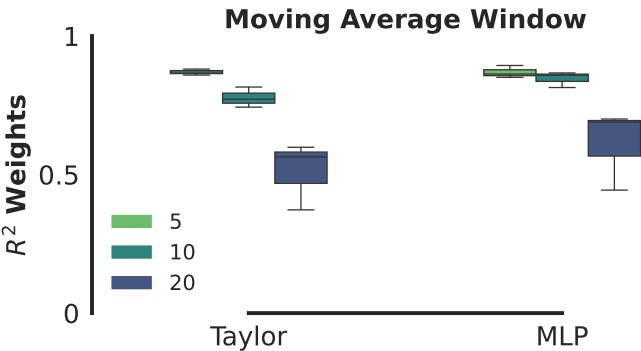

Figure 6: Effect of moving average window (used for calculated expected reward) on the performance of learned plasticity rule

### A.5.3 Input firing mean

This is the firing mean encoding used for representing an odor. For example, a firing mean of 1 represents the odor as [1, 0], with additional Gaussian white noise fixed at a variance of 0.05. Our model (specifically the MLP implementation) is able to account for a range of input firing means (Fig. 7).

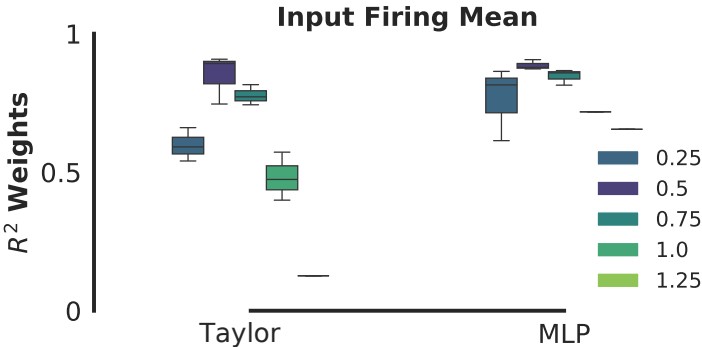

Figure 7: Effect of input firing mean (used for odor representation) on the performance of learned plasticity rule

### A.6 Validation on held-out data

We ran a validation experiment to assess the extent to which the fitting procedure could generalize to unseen biological data. Due to the sequential nature of the dataset, and that we are fitting our model to individual flies' behavioral trajectories (*i.e.*, sequences of choices), we cannot perform classical $k$-fold cross-validation in which a random subset of timepoints or trials are held out. Instead, we train the model using the first $x\%$ of an individual fly's trajectory, and then we test it on the last $(100 - x)\%$ (Figure 8). To ensure there is no data leakage from the training set, we re-initialize the model's synaptic weights at the beginning of each test sequence, although the test performance is similar if the initial weights are carried over from the final timestep of the training sequence.

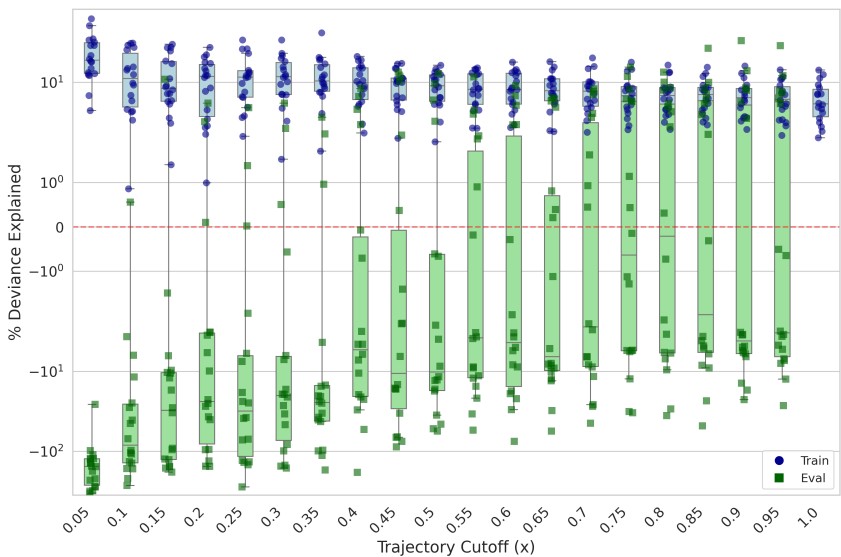

Figure 8: Percent deviance explained on the training and test data, training on $x\%$ of the fly data trajectory and testing on the remaining $100 - x\%$.

Interestingly, we found that some flies had test performance close to training performance, whereas others had poor test performance (Figure 8). The flies with good test performance (defined as having a positive percent deviance explained on the test set) reaffirmed our conclusion that a weight decay term enabled a quantitatively better description of fly behavior (Figure 9, top). The flies with poor test performance (negative test percent deviance explained) also had negative plasticity coefficients for the $w$ term ($\theta_{001}$ in Equation 6), suggesting that the differences in test performance were not due to different estimates of this coefficient (Figure 9, bottom). More analyses are required to determine why some flies had markedly better test performance than others.

### A.7 Additional plasticity rules

In these additional experiments, we maintain the reward term as the difference from the expected reward. This approach facilitates bidirectional plasticity. Additionally, we incorporate a weight decay term, experimenting with several coefficients, ultimately choosing a value of 0.05 as it seems reasonable for our experimental configuration. Table 3 presents a comparison between the MLP and the Taylor series, reporting the $R^2$ over weights, $R^2$ over activity, and the percent deviance explained. At a high level, both methods appear to perform similarly. To gain deeper insights into why certain rules are more "recoverable" than others, an examination of weight dynamics for each method is necessary. The Taylor series model has 81 trainable parameters, while the MLP has 61. Fitting the rules with a relevant subset of the Taylor series, selected through biological priors as done in the *Drosophila* experimental data, is expected to result in better performance.

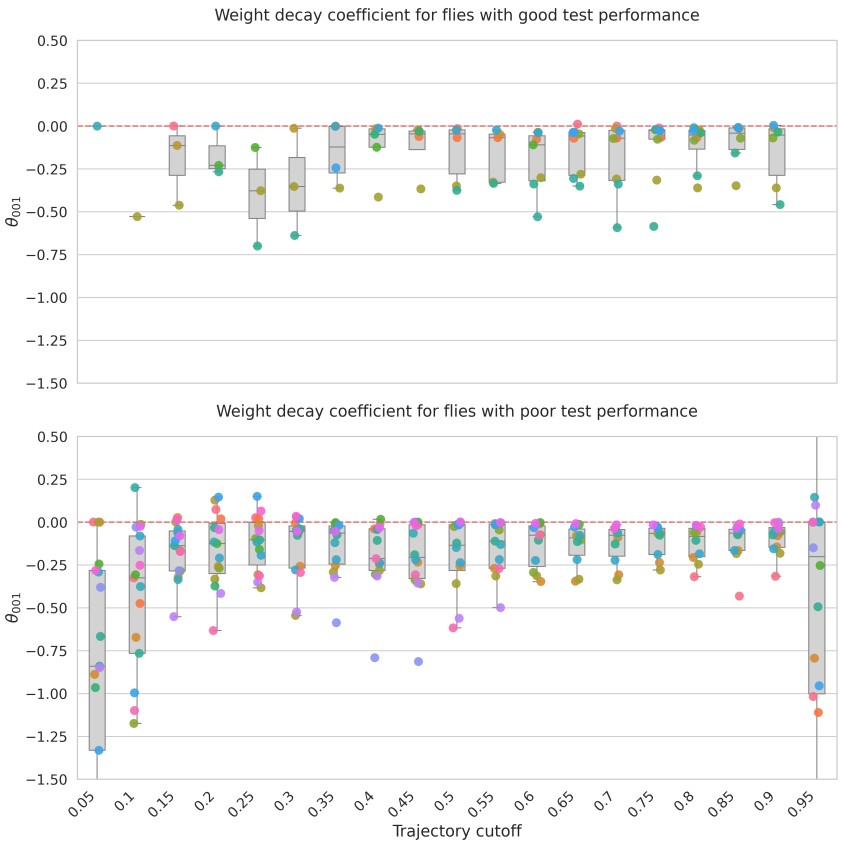

Figure 9: Learned weight decay term in the Taylor series parametrized plasticity rule (Equation 6), from flies with positive (top) and negative (bottom) test set percent deviance explained.

Table 3: Evaluation of various different reward-based plasticity rules: $R^2$ scores for weight and individual neural activity trajectories, and percentage of deviance explained for behavior.

| Plasticity Rule | MLP | | | Taylor | | |
|---|---|---|---|---|---|---|
| | $R^2$ Weights | $R^2$ Activity | % Deviance | $R^2$ Weights | $R^2$ Activity | % Deviance |
| $x_j y_i w_{ij} r - 0.05r$ | 0.27 | 0.34 | 79.92 | 0.41 | 0.51 | 84.22 |
| $x_j w_{ij} y_i r - 0.05r$ | 0.61 | 0.67 | 84.39 | 0.33 | 0.30 | 82.79 |
| $x_j w_{ij} r^2 - 0.05r$ | 0.66 | 0.66 | 84.75 | 0.42 | 0.47 | 79.33 |
| $x_j y_i w_{ij} r^2 - 0.05r$ | 0.40 | 0.57 | 80.56 | 0.40 | 0.48 | 79.03 |
| $x_j r^2 - 0.05r$ | 0.75 | 0.73 | 70.78 | 0.66 | 0.61 | 70.65 |
| $x_j^2 y_i w_{ij} r - 0.05$ | 0.59 | 0.68 | 70.41 | 0.59 | 0.69 | 68.84 |
| $x_j^2 y_i^2 r^2 - 0.05$ | 0.59 | 0.62 | 64.98 | 0.55 | 0.62 | 64.44 |
| $x_j r - 0.05 x_j y_i r$ | 0.81 | 0.96 | 63.64 | 0.81 | 0.95 | 64.01 |
| $x_j r - 0.05 x_j w_{ij} r$ | 0.84 | 0.95 | 64.19 | 0.82 | 0.95 | 63.24 |
| $x_j r - 0.05 x_j y_i$ | 0.80 | 0.96 | 62.90 | 0.76 | 0.94 | 62.50 |
| $x_j r$ | 0.85 | 0.96 | 64.76 | 0.78 | 0.94 | 61.91 |
| $x_j r - 0.05r$ | 0.70 | 0.76 | 62.01 | 0.63 | 0.68 | 61.82 |
| $x_j r - 0.05 x_j y_i w_{ij} r$ | 0.82 | 0.96 | 63.13 | 0.73 | 0.93 | 61.05 |
| $x_j r - 0.05 x_j w_{ij}$ | 0.83 | 0.95 | 61.24 | 0.81 | 0.95 | 60.85 |
| $x_j r - 0.05 x_j y_i w_{ij}$ | 0.84 | 0.96 | 63.53 | 0.72 | 0.92 | 60.33 |
| $x_j r - 0.05 x_j$ | 0.89 | 0.97 | 61.99 | 0.77 | 0.91 | 60.05 |
| $x_j r^2 - 0.05 x_j w_{ij}$ | 0.89 | 0.95 | 54.92 | 0.88 | 0.94 | 53.94 |
| $x_j r^2 - 0.05 x_j y_i$ | 0.79 | 0.95 | 56.40 | 0.77 | 0.92 | 53.76 |
| $x_j r^2$ | 0.89 | 0.96 | 54.32 | 0.81 | 0.92 | 52.87 |
| $x_j r - 0.05 w_{ij}$ | 0.87 | 0.91 | 57.01 | 0.70 | 0.86 | 51.80 |
| $x_j r^2 - 0.05 x_j y_i r$ | 0.84 | 0.96 | 53.14 | 0.78 | 0.93 | 51.61 |
| $x_j r^2 - 0.05 x_j y_i w_{ij}$ | 0.81 | 0.95 | 54.92 | 0.76 | 0.93 | 51.52 |
| $x_j r^2 - 0.05 x_j w_{ij} r$ | 0.85 | 0.96 | 53.04 | 0.78 | 0.92 | 51.30 |
| $x_j r^2 - 0.05 x_j y_i w_{ij} r$ | 0.90 | 0.96 | 53.90 | 0.82 | 0.92 | 51.29 |
| $x_j r^2 - 0.05 x_j$ | 0.89 | 0.95 | 51.52 | 0.85 | 0.93 | 51.03 |
| $x_j r^2 - 0.05 x_j r$ | 0.88 | 0.96 | 54.17 | 0.79 | 0.93 | 50.90 |
| $x_j r - 0.05 y_i w_{ij} r$ | 0.91 | 0.95 | 52.94 | 0.84 | 0.91 | 50.24 |
| $x_j r - 0.05 y_i r$ | 0.94 | 0.95 | 52.84 | 0.86 | 0.92 | 50.20 |
| $x_j r - 0.05 w_{ij} r$ | 0.87 | 0.94 | 49.72 | 0.82 | 0.92 | 48.90 |
| $x_j r - 0.05 y_i w_{ij}$ | 0.94 | 0.96 | 48.58 | 0.90 | 0.95 | 48.60 |
| $y_i w_{ij} r^2 - 0.05$ | 0.66 | 0.90 | 47.26 | 0.55 | 0.84 | 47.00 |
| $x_j r^2 - 0.05 w_{ij}$ | 0.86 | 0.94 | 46.44 | 0.78 | 0.91 | 46.26 |
| $x_j r^2 - 0.05 y_i w_{ij} r$ | 0.91 | 0.92 | 43.64 | 0.89 | 0.93 | 44.04 |
| $x_j r^2 - 0.05 y_i r$ | 0.96 | 0.97 | 43.74 | 0.93 | 0.95 | 43.18 |
| $x_j r^2 - 0.05 w_{ij}$ | 0.92 | 0.93 | 42.68 | 0.91 | 0.94 | 42.43 |
| $x_j^2 y_i w_{ij} r^2 - 0.05r$ | 0.65 | 0.53 | 43.26 | 0.65 | 0.57 | 42.32 |
| $x_j r^2 - 0.05 y_i r$ | 0.93 | 0.92 | 44.13 | 0.91 | 0.94 | 41.69 |
| $y_i w_{ij} r - 0.05$ | 0.69 | 0.89 | 38.52 | 0.61 | 0.85 | 37.74 |
| $y_i^2 r^2 - 0.05$ | 0.52 | 0.87 | 37.01 | 0.60 | 0.82 | 36.81 |
| $x_j r^2 - 0.05 y_i$ | 0.97 | 0.97 | 34.55 | 0.96 | 0.97 | 34.36 |
| $x_j^2 y_i^2 r^2 - 0.05r$ | 0.72 | 0.65 | 33.94 | 0.68 | 0.60 | 33.57 |
| $x_j^2 y_i w_{ij} r - 0.05r$ | 0.60 | 0.53 | 34.82 | 0.65 | 0.57 | 33.45 |
| $y_i w_{ij} r - 0.05 x_j r$ | 0.04 | 0.45 | 28.56 | 0.08 | 0.46 | 28.63 |
| $x_j^2 y_i^2 r - 0.05r$ | 0.50 | 0.44 | 28.81 | 0.70 | 0.60 | 28.08 |
| $y_i w_{ij} r^2 - 0.05 x_j r$ | 0.17 | 0.47 | 26.90 | 0.31 | 0.44 | 24.92 |
| $y_i w_{ij} r^2 - 0.05r$ | 0.38 | 0.32 | 16.49 | 0.60 | 0.59 | 15.87 |
| $y_i^2 r^2 - 0.05r$ | 0.01 | −0.01 | 14.06 | 0.24 | 0.29 | 13.62 |
| $x_j^2 y_i^2 w_{ij} r - 0.05 x_j r$ | 0.82 | 0.94 | 4.64 | 0.86 | 0.95 | 4.35 |
| $x_j^2 y_i^2 w_{ij} r^2 - 0.05 x_j r$ | 0.78 | 0.94 | 4.31 | 0.90 | 0.96 | 4.01 |
| $x_j r - 0.05 x_j r$ | 0.75 | 0.92 | 4.30 | 0.92 | 0.96 | 3.96 |
| $x_j y_i^2 w_{ij} r^2 - 0.05 x_j r$ | 0.79 | 0.93 | 4.39 | 0.89 | 0.97 | 3.87 |
| $x_j y_i w_{ij} r - 0.05 x_j r$ | 0.70 | 0.91 | 3.49 | 0.87 | 0.96 | 3.29 |
| $x_j y_i^2 r - 0.05 x_j r$ | 0.63 | 0.90 | 3.28 | 0.83 | 0.96 | 3.04 |

