# OpenReview forum: "Model Based Inference of Synaptic Plasticity Rules"
_NeurIPS.cc/2024/Conference — NeurIPS 2024 poster_

### Official Review · Reviewer_MyoH · 2024-07-08

**Soundness:** 4
**Presentation:** 4
**Contribution:** 3
**Rating:** 7
**Confidence:** 4

**Summary:**

This paper proposes a novel method for inferring plasticity rules from neural and behavioral data. In contrast to previous approaches, the plasticity rule is directly optimized to maximize the similarity of the output of a model trained with the plasticity rule to a target (neural activity or behavior). This approach is validated with synthetic experiments and used to infer plasticity rules from fruit fly behavioral data.

**Strengths:**

* Determining the functional roles of synaptic plasticity is a fundamental problem in computational & theoretical neuroscience.
* The method is not particularly novel, but it is simple, nicely explained, flexible, and well-validated.
* The paper is very well-written, and all methods and findings are clearly described.

**Weaknesses:**

* My sole concern with this method is its broader applicability to more complex problems; inferring the learning dynamics involved in a simple forced choice task where the biological network is relatively well-known is very different from partially observed neural activity with poorly-understood connectivity during free behavior. I'm particularly concerned about the case where the influence of a large number of unobserved neurons needs to be considered.

**Questions:**

* Double-check the subscripts in Eq (5), I think $y_j$ should be $y_i$.
* For the bottom portion of Fig 2, are the weights being compared the weights after training when using the ground truth and inferred rules?
* Is there some explanation for why the last rule in Table 1 exhibits such different behavior?
* Is there held out validation data in the experiments in Section 5, or is a single dataset used throughout?

**Limitations:**

Yes

---

> ### Author Rebuttal · Authors · 2024-08-07
>
> # Response to reviewer MyoH
>
> Dear Reviewer,
>
> Thank you for your feedback and your recognition of the interesting aspects of our paper. We have addressed your highlighted Weaknesses and Questions one by one below:
>
> ## Weaknesses
> 1. We acknowledge that when choosing problems to test our approach we used behavioral settings and plasticity rules that were relatively simple. Our choice of problems was inspired by the kinds of behavioral data that our colleagues had available to test our approach on, and it is something that can be expanded upon in the future to include more complex problems. Despite this, we believe that the experiments described in this paper suggest that our approach can in fact be applied to more complex situations. Our reasoning for this is two-fold:
>     - First, while we made the theoretical assumption  in this paper that the plasticity rule is a function of only presynaptic activity, postsynaptic activity, reward, and the current weight value, our fitting procedures are designed such that additional terms can be added without difficulty. This would allow us to estimate a large variety of plasticity rules.
>     - Second, we simulated and were able to retrieve a large number of basically arbitrary plasticity rules (in the Appendix) from neural trajectories. This estimation also showed resilience to noise and sparsity of the input data.
>
> ## Questions
> 1. You are correct that  the subscripts in Eq (5) had a typographical error. We have updated the text to reflect this. Thank you for pointing this out.
>
> 2. Regarding the R-squared calculation in Figures 2E and 2F, we calculated the R-squared score by comparing the weight trajectories of the ground truth rule with the weight trajectories of the inferred rule after the learning rule optimization was completed. In particular, our R-squared comparison was not limited to the final weights, and it instead compared the full trajectories.
>
> 3. The Reviewer is correct that our method sometimes infers plasticity rules that poorly approximate the weight trajectories, activity trajectories, and/or behavioral trajectories (Tables 1 and 2). This isn’t surprising because plasticity rules are nonlinear dynamical systems, and their predictions may depend very sensitively on parameters and initial conditions. We would contend that no inference method will ever be able to perfectly recover an arbitrary plasticity rule, and the intention of Tables 1 and 2 is to merely illustrate the range of things that are possible. Nevertheless, we would hypothesize that the rules implemented by biology are less unwieldy than some of the possibilities revealed by the tables. Indeed, our method works very reliably on the canonical rules that were designed to do useful computations and/or capture biological phenomena. We hope that using our framework to fit real data will help us to identify novel, well-behaved rules without necessitating laborious hand design.
>
> 4. In Section 5, we fit behavioral data from 18 flies, utilizing the full dataset for our reported results. We acknowledge, however, that incorporating cross-validation would strengthen our analysis. If our paper is accepted for publication, then we commit to implementing this approach and including it in the final version of our manuscript, thereby enhancing the robustness of our model evaluation. It's worth noting that in our simulation experiments, we employ a separate held-out test set to assess model performance.

---

> > ### Comment · Reviewer_MyoH · 2024-08-08
> >
> > I appreciate the thoughtful response. I maintain that this is an interesting paper that proposes a clean and fairly general method to an important problem and recommend it for acceptance on that basis alone. I agree with the other reviewers that the potential for broader impact is less clear. There seem to be good reasons to doubt that this method or its underlying principles will scale to more complex paradigms, but this is ultimately an empirical question which will only be answered by substantially more involved experiments in follow-up work. I stand by my original score.

---

### Official Review · Reviewer_5Cze · 2024-07-12

**Soundness:** 3
**Presentation:** 4
**Contribution:** 3
**Rating:** 7
**Confidence:** 3

**Summary:**

This paper studies how to learn local learning rules (like those plasticity rules thought to be used by real neurons) in a data-driven way from neural activity or behavioral timeseries. This is applied to simulated as well as fly behavioral data. The true learning rules can be learned from synthetic activity timeseries when complete observations are available (Fig 2), whereas from behavior alone some more error is incurred (Fig 3). When using fly behavioral data, the authors get better fits than previous methods (Fig 4), and a claim is made that the "forgetting term" in the resulting fit is important.

**Strengths:**

The paper studies an interesting inference problem which is relevant to both AI and neuroscience. It is well-written with a discussion of some limitations. It studies both synthetic and real data. I found the idea of trying to fit this kind of model to behavior interesting. I wouldn't have thought that it would be possible, to be honest.

**Weaknesses:**

I think the main weakness is what I identified below in limitations: the degeneracy or non-identifiability of such models. I can imagine that even in settings where the equations are the correct model that it is not identifiable without some regularization.

The model fits to behavioral data are given as R^2 values on the behavior themselves. It does not seem that any cross-validation was used for model fitting, so there is a possibility that your model is overfitting. This should be discussed.

I'll admit I'm not super familiar with existing work in this area, so I am not sure how novel these results are relative to what's been tried before.

**Questions:**

* What justification do you have for claiming that your model requires less energy than an LLM of the same size? (Line 45-46)
* Can you be precise about which term in Fig 4C is the decay term? I think you mean 000. It isn't evident whether that is significantly different from 0. The error bars are quite wide.
* Have the authors considered the vast literature on fitting dynamics themselves from timeseries? I am familiar with the SINDy method, which uses polynomial or other families of basis functions and includes a LASSO-type penalty to fit dynamics. It is likely that some regularization or penalty could improve the learned models in your setting, too. (I now see this buried in the appendix, but it should be discussed in the main text.)
* Please discuss why you did or didn't use cross-validation

**Limitations:**

I think it's a stretch to claim that the underlying connectivity (presence of nonzeros in weight matrix) is known because of connectomics (lines 80-82). That may be true for some small model organisms like Drosophila, and only just, but it is nowhere near true for others such as the mouse. The authors should be careful about making claims of identifiability with models such as theirs. For instance, it's known that incomplete knowledge of the connectome or partial observations of dynamics can lead to spurious correlations. An example that comes to mind is https://journals.aps.org/pre/abstract/10.1103/PhysRevE.109.044404 .

Your experiments actually show this non-identifiability already: In figure 2G we see that the Oja rule is not recovered when the observations are incomplete. There is a brief discussion of this "degeneracy" in the conclusions and limitations section, but I think it is a bigger issue than the authors are claiming. In particular, this conjecture that "in the infinite data limit the fitted plasticity rules would, in fact, be unique" could easily be wrong. It is certainly not a useful argument to the experimentalist who will always have limited data, especially if it's behavioral or brain recording data.

---

> ### Author Rebuttal · Authors · 2024-08-07
>
> # Response to reviewer 5Cze
> Dear Reviewer,
>
> Thank you for your careful reading of our work and insightful comments. We are glad that you found the paper to be “interesting,” and “relevant.” We address each of the comments mentioned in your Weaknesses, Questions, and Limitations sections below.
>
> ## Weaknesses
> 1. We have tried to distinguish between a model being non-identifiable, by which we mean that its parameters are impossible to uniquely infer, and sloppy, by which we mean that its parameters are poorly constrained by available data.
>
> 2. Regarding sloppiness, we agree that it could be hard to estimate the parameters of the plasticity rule in practice given finite data sets. Multiple terms in the plasticity rule may lead to similar weight changes. Nevertheless, our paper shows that it is empirically possible to infer plasticity rules given reasonable amounts of simulated data. It also identifies a biologically interpretable but previously unknown term in the plasticity rule of the fly mushroom body. This term would not appear consistently in the statistical fits of individual flies if it was too sloppy to be determined. Simply, our method is already useful, regardless of whether it can detect the sloppiest components of plasticity rules.
>
> 3. Regarding identifiability, if one could measure every term that enters the plasticity rule, then the identifiability of the plasticity rule would follow from the uniqueness of Taylor series expansions. Since weights are assumed to be unmeasured, then it’s in principle possible that some plasticity rules may be non-identifiable. This seems unlikely to us, because each component of the plasticity rule contributes to weight changes that measurably affect the postsynaptic activity. Nevertheless, we acknowledge that we don’t have a formal mathematical proof that the plasticity rule is identifiable with unmeasured synaptic weights.
>
> 4. In our opinion, identifiability is less relevant than sloppiness, and sloppiness is less relevant than our empirical demonstrations. Our method already works well enough to be applied fruitfully to simulated and real-world problems.
>
> 5. We acknowledge the reviewer's concern regarding the potential for overfitting due to the absence of cross-validation. We want to clarify that in our simulated experiments we consistently utilized a separate held-out test set to evaluate model performance - we have clarified this in the main text. However, we recognize the value of incorporating cross-validation for fitting the fly experimental data. If the paper is accepted for publication, we will implement a cross-validation procedure and include it in the final version of our manuscript. We agree with the reviewer that this would substantially enhance the robustness of our model evaluation.
>
> ## Questions
> 1. We appreciate the reviewer's observation and have removed the statement about energy implications for training LLMs, as upon further consideration the point seemed tangential to the goals of our paper.
>
> 2. We have added text to the caption of Fig 4C to clarify that parameter 001 corresponds to the weight decay term, and have listed what each of the terms mean. The 000 term is a bias.
>
> 3. We thank the reviewer for pointing us to the literature that focuses on fitting dynamical models from time-series data, especially the idea that regularization penalties can substantially improve model performance. As the reviewer pointed out, we have begun exploring the L1 regularization penalty in our model, and we are familiar with the related SINDy approach. If accepted, we will incorporate a more thorough investigation of the impact of regularization on model performance, and we will move the relevant text discussing this from the Appendix to the main manuscript.
>
> 4. We have already addressed this concern in the “Weaknesses” section of our response.
>
>
> ## Limitations
>
> 1. The reviewer’s comment regarding connectomics is addressed in the general response to all reviewers.
>
> 2. We have already discussed identifiability and sloppiness in our response to the Reviewer’s “Weaknesses” comments. Here the Reviewer additionally brings up “model mismatch,” by which we mean that best-fit model parameters are not directly interpretable if the model parameters are out of accordance with the underlying biology. We agree that this is very often an issue with model fitting, and we certainly don’t claim to have a solution to this general problem. We don’t currently see anything that we could add to the study that would alleviate this concern, and our approach has instead been to illustrate related failure modes. Indeed, the Reviewer points to our figures to illustrate the point, which we think indicates that we’ve given it fair treatment. We are happy to make additional revisions if the Reviewer could clarify what they find problematic about our treatment. The quoted text that the Reviewer provides is about “identifiability,” not “model mismatch,” and we think it is important to keep the related issues of identifiability, sloppiness, and model mismatch distinct as these issues are fundamentally different.

---

> > ### Comment · Reviewer_5Cze · 2024-08-11
> >
> > Thanks for your responses. I think that if you address these comments in sufficient detail in the final version, that is sufficient. I am going to adjust my overall score to a 7 and soundness score to a 3.

---

### Official Review · Reviewer_2Jtn · 2024-07-20

**Soundness:** 3
**Presentation:** 3
**Contribution:** 4
**Rating:** 7
**Confidence:** 4

**Summary:**

This paper presents a reward learning-based method for recovering biological synaptic plasticity rules in a model network. It is meant to be applied to both neural and behavioral data. The method consists of taking real learning trajectories collected in response to stimuli (of some observable metric modality) and showing those stimuli to the model artificial neural network, then computing the loss between the generated and. modeled trajectories and taking a gradient step in the direction of the ground truth data. The weight update is determined by some plasticity function, represented by either a Taylor expansion or an MLP, and also learned over the course of training according to the trajectory loss. After training the model network, various plasticity rules are tested by analyzing the learned parameter values of the plasticity function.

After establishing the method, the paper presents specific experimental settings and results. It begins with a simpler case of simulated neural activation data with plasticity dynamics based on Oja's rule, MSE loss between the two trajectories, and a Taylor series plasticity function. The paper shows that Oja's rule can be recovered - it is seen clearly in the learned parameters of the Taylor series plasticity function. Ablations are also conducted over increasing sparsity and noise, showing degradation.

The paper then goes on to a simulated behavioral data setting with a small MLP representing the plasticity function. Here, it establishes percent deviance explained between the ground truth and modeled trajectories as a performance metric. Finally, it demonstrates some proof of concept in actual drosophila behavioral data, including a forgetting mechanism (negative dependency on the weights, not just the reward signal and presynaptic activations).

Finally, it goes through related work.

**Strengths:**

### Originality
As far as I am aware, this particular reward-based training set up with an organism model and a separately trained plasticity model is novel.

### Significance
If viable, this would be a very useful work for modeling. This is an initial exploration, but the significance of the results is nontrivial - it opens up a research direction.

### Quality
- Training setup is creative! It would be nice to know how well this scales, as trajectory-based sequence learning is difficult.
- Experimental settings are thoughtful and elucidating
  - The toy example does an especially good job of helping the reader build intuition
  - Showing that synaptic trajectory error improves over time even though training is done with neuronal trajectories - this is clever and convincing
  - Real drosophila experiment is exciting
- Analysis of plasticity functions, how they show up in parametrization, and what that means, is very itneresting

### Clarity
- Results has a claims-driven structure that is helpful for understanding takeaways
- Methods are very clear

**Weaknesses:**

### Clarity
- Paper is very jargon-heavy without defining niche terms and having clear conclusions in paragraphs. It would help to have simpler language and/or more explanation
- Percent deviance explained - we don't get grounding or baselines to understand how to judge these results. Are there baseline methods, or any kind of comparison point/context?

### Quality
- It's not clear how far this method, or even the principles established in the paper, can take us.
  - The space of tested plasticity rules seems hand-designed and limited. Rules are rejected only when they can be clearly defined and tested.
  - Percent deviance explained results seem to leave plenty of room for improvement
  - Test setups are quite simple. This is mitigated highly by the presence of real drosophila behavior data - that experiment is really useful - but still, simple and limited, particularly in the underlying dynamics
  - The paper argues that it is "reasonable" to assume that we can have the modeling network architecture exactly match the true architecture, because of available connectome data. Connectomes are static and extremely detailed, and there are lots of elements (e.g. immense recurrence) that we do not have architectures for. This is again very much improved by the real drosophila data experiment, but it is nevertheless concerning.
  - Underlying plasticity rules in synthetic settings are very simple, even in the case being learned by an MLP. The drosophila behavioral experiment helps because we see the discovery of the forgetting rule, but we don't have any sense of a full space to look for, or even a partial but extensive space to look for (things that both should and shouldn't show up). Even the "forgetting" rule discovery is believable but requires a lot of interpretation of results.
- Ablations aren't really discussed. Noise and sparsity make sense as ablation factors, but what do the results mean beyond just "noise and sparsity cause problems"? Especially because their effects are strangely similar.

**Questions:**

- How should I ground my judgement of the percent deviance explained results?

**Limitations:**

Limitations - yes
Societal impact - they include the NeurIPS checklist in their supplementary and say the societal impacts have been addressed, but they haven't anywhere, nor is there an assertion that they don't need to. They probably don't, though.

---

> ### Author Rebuttal · Authors · 2024-08-07
>
> # Response to reviewer 2Jtn
>
> Thank you for your constructive feedback on our manuscript. We are glad that you found our work to be “creative,”  and “clever and convincing.” Below, we address each of the concerns you expressed in your review, and hope that you will find our revised manuscript to be improved.
>
> ## Clarity
>
> 1. We have added conclusions to paragraphs where this was missing and simplified language that was  jargon-filled. These changes are highlighted in red. To help readers understand any remaining specialized language, we added a Glossary before the Appendix. If the Reviewer alerts us to additional terms that they found to be jargon, we would be happy to either eliminate or add it to our Glossary depending on its utility.
>
> 2. (Also the response for Question 1) One should interpret the percent deviance explained similarly to the R-squared metric. A value of 0% corresponds to chance performance and a value of 100% corresponds to theoretically optimal performance. It’s useful to our paper because it doesn’t assume continuous outputs or Gaussian noise. It thus applies to binary behavior more naturally than the R-squared metric does. A definition of this metric is provided in Appendix section A3 and we have added a reference to this in the text. We apologize that this reference previously pointed to an incorrect section of the paper.
>
> ## Quality
>
> 1. We agree that the space of plasticity rules is hard to specify and sample in its entirety. We’d like to clarify two related issues.
>     - First, we acknowledge that when choosing plasticity rules to simulate ground-truth data, we emphasized canonical models that have been “hand-designed” by the computational neuroscience community due to their  properties and biological relevance. Their simplicity reflects the current state of understanding in the field. However, we also simulated a large number of other rules in the Appendix (Table 2) excluding the unstable rules. We chose this naive sampling of plasticity rules because the field does not yet have the knowledge to know what class of rules are most relevant. Given this, a better approach was not clear to us as that the space of simulated rules cannot be sampled exhaustively.
>
>     - Second, we’d like to note  that our framework and fitting procedures were designed with the goal of leveraging large-scale biological data to help move beyond hand-designed rules. Our paper’s main assumption is that the plasticity rule is a function of only presynaptic activity, postsynaptic activity, reward, and the current weight value. Our fitting procedures are designed to estimate this unknown function from data. There are multiple ways to parameterize functions, and we consider two parameterizations with complementary strengths. First, we use a low-order polynomial, which can be taken as estimating the Taylor series for the function and is easy to interpret as it relates simply to the canonical rules. Second, we use a multilayer perceptron, which sacrifices interpretability for expressivity. In either case, we’re providing a general parameterization that can reveal unexpected results when fit to biological data.
>
> 2. The Reviewer is correct that our method sometimes infers plasticity rules that poorly approximate the weight, activity, and/or behavioral trajectories (Table 2). This isn’t surprising because plasticity rules are nonlinear dynamical systems, and their predictions may show sensitivity to parameters and initial conditions. We would contend that no inference method will be able to perfectly recover an arbitrary plasticity rule. The intention of Table 2 is to merely illustrate the range of possibilities. Nevertheless, we hypothesize that the rules implemented by biology are less unwieldy than some possibilities in Table 2. Indeed, our method works very reliably on the canonical rules that were designed to capture biological phenomena. We hope that using our framework to fit real data will help us to identify novel, well-behaved rules without necessitating laborious hand design.
>
>
> 3. As explained above, we acknowledge that we’ve emphasized tests against simple, plausible learning rules. However, the suite of learning rules in Table 2 include a wide range of ground-truth plasticity rules with nonlinear (polynomial) dependencies among the terms. We agree that applying our method to real data is a productive path forward, and found it very encouraging that the method already uncovered something novel and interpretable in the Drosophila data. We anticipate more discoveries with richer datasets and models.
>
> 4. The reviewer’s comment regarding connectomics is addressed in the general response to reviewers.
>
> 5. We believe this comment on the simplicity of the synthetic rules has been adequately addressed by our earlier responses.
>
> 6. We have added text to Section 3.1 to clarify the interpretation of our choice of ablation experiments. This was guided by the kinds of incomplete information we would likely deal with when applying our model to biological data. Often neural recordings are noisy and even the most state-of-the-art recording tools  suffer from not being able to record entire populations of interest. Our ablation experiments quantify how resilient our approach is to these sources of error that are often present in real data.
>
> ## Limitations
>
> 1. We have changed our response to the societal impact question of the NeurIPs checklist and provide a justification for why this is not discussed in the main text. While we have discussed our paper’s impact within the field of neuroscience, the larger societal impact will likely only be revealed as the method is used more widely to understand specific neural systems and their plasticity rules. As an immediate societal impact is exceedingly unlikely, we did not dedicate a section to discuss these issues.
>
> 2. A description about the scalability of our approach has been provided in the general rebuttal to all reviewers.

---

> > ### Comment · Reviewer_2Jtn · 2024-08-13
> > **Increasing score to 7**
> >
> > Thanks for the detailed comments. The main thing imo is that you've successfully argued that the limitations of the paper are more reasonable and field-standard than I was aware of. Increasing my score to a 7.

---

### Author Rebuttal · Authors · 2024-08-07

# General response to reviewers

We appreciate the thoughtful and constructive feedback provided by all reviewers. We have revised our manuscript based on this. This rebuttal addresses two key points relevant to all reviewers.

## **1. Scalability of the method**

In response to Reviewer 2Jtn's interest in knowing "how well the method scales as trajectory based learning sequence learning is difficult", we share the following hot-off-the-press *preliminary* results. Our primary results (Figures 3, 4, Table 1, and supplementary materials) used a trajectory length of 240 and a [2-10-1] neural network architecture. To address scalability, we conducted additional analyses varying both trajectory length and hidden layer size. These analyses used simulated data with a ground truth rule of x.(r - E(r)) on behavioral data, employing a Taylor series plasticity function parameterization. Results were averaged over three seeds for robustness.

It's noteworthy that for a [2-1000-1] architecture, our method performs backpropagation through time over 2000 synapses across 240 time points, with each synapse updated at every time point following the parameterized Taylor expansion. This demonstrates that computational complexity increases with both network size and trajectory length.

### **A) Scalability with trajectory length**

| Trajectory length | 30   | 60   | 120  | 240  | 480  | 960  | 1920 |
|-------------------|------|------|------|------|------|------|------|
| R2 Activity       | 0.92 | 0.91 | 0.91 | 0.94 | 0.95 | 0.87 | 0.92 |
| R2 Weights        | 0.79 | 0.74 | 0.70 | 0.78 | 0.72 | 0.53 | 0.64 |
| Percent Deviance  | 39.52| 40.14| 48.06| 61.91| 76.90| 73.78| 79.66|

The model's goodness-of-fit generally improved with longer simulations, likely due to more data points for inferring the plasticity rule. However, R-squared values for activity and weights peaked before declining, suggesting potential overfitting on very long trajectories. We plan to add cross-validation analysis in the final manuscript if accepted. The current trajectory length (240) appears near optimal for R-squared values, mitigating overfitting concerns for the main results.

### **B) Scalability with network architecture**

Our primary findings use a [2-10-1] architecture (20 synapses updated at every time point). We've demonstrated that the framework scales to 1000 hidden units (2000 synapses).

| Hidden Layer Size | 10    | 50    | 100   | 500   | 1000  |
|-------------------|-------|-------|-------|-------|-------|
| R2 Activity       | 0.94  | 0.94  | 0.95  | 0.95  | 0.95  |
| R2 Weights        | 0.78  | 0.75  | 0.79  | 0.79  | 0.79  |
| Percent Deviance  | 61.91 | 62.29 | 62.25 | 62.27 | 62.26 |

Model performance remains consistent when scaling to larger synapse counts, assuming the same plasticity rule is applied.

We are currently conducting additional experiments on scalability with respect to the parameters in the plasticity function, focusing on:
1. Plasticity MLP complexity (size)
2. Number of terms in the Taylor expansion (up to 3^4)

These findings will be included in the camera-ready version of the manuscript, if accepted.

## **2. Connectomics and model architecture**

We acknowledge the reviewers' correct observation that connectomes miss important architectural information and are most complete only in small model organisms. We will revise the text to avoid overstating the relevance of this data for larger brains.

Despite these limitations, several research groups have successfully used connectomics to build biologically realistic network models by parameterizing and fitting the most important unknown quantities [1-3]. For example, while connectomes provide synapse counts but not weights, accurate neural network models have been built by fitting cell-type-specific scale factors that convert synapse counts to weights.

Dynamical neural network models could be similarly designed to include synaptic plasticity:
1. The connectome could be used to infer an adjacency matrix at the level of cell types or single cells.
2. Plasticity rules could be parameterized, as in this paper, and fit to neural/behavioral dynamics.
3. The probability distribution of synapse counts could be used to constrain the probability distribution of synaptic weights, providing constraints on the plasticity rule.

We have two main reasons for assuming it's reasonable to match the architecture to connectomics data:

1. In Section 3, our intention was to demonstrate that our approach can solve the inference problem in a setting where the predictive and generative network architectures matched. This scenario is indeed aspired to by fly neuroscientists in the era of connectomics. As the reviewers point out, we move away from this assumption when modeling the behavioral data in Sections 4 and 5.

2. We argue that the available connectomic information is sufficiently rich to allow us to construct neural networks similar to the structure of brain regions of interest. In Section 5, we show that such a mushroom-body-inspired architecture allows us to infer rules that agree with predominant ideas in the field and contribute unique and interpretable additions to the knowledge base.

We have added language to clarify that while we believe using connectomic information to design model architecture is reasonable, mismatches in generative and model architectures can lead to errors in interpretation. These should be taken into consideration, and we have added a citation to the reference suggested by reviewer 5CZe.

**References**

[1] Lappalainen et al. *Connectome-constrained deep mechanistic networks predict neural responses across the fly visual system at single-neuron resolution*, (bioRxiv, 2023)

[2] Mi et al. *Connectome-constrained Latent Variable Model of Whole-Brain Neural Activity*, (ICLR, 2021)

[3] Beiran et al. *Prediction of neural activity in connectome-constrained recurrent networks*, (bioRxiv, 2024)

---

### Decision · Program_Chairs · 2024-09-25

**Decision:**

Accept (poster)

**Comment:**

The paper proposes a nice approach for inferring synaptic plasticity rules from (partially) observed neural activity or behavioral read-outs. The approach is fairly straightforward, but it seems to work well across several examples. I found it surprising and impressive that behavior in a binary decision making task could say much about the nature of the plasticity rule.

The reviewers raised some very good points, and I encourage the authors to incorporate their feedback and make the promised changes.